# Towards Stable Representations for Protein Interface Prediction

**Ziqi Gao[1,2], Zijing Liu[3*], Yu Li[3], Jia Li[1,2*]**
[1]Hong Kong University of Science and Technology
[2]Hong Kong University of Science and Technology (Guangzhou)
[3] International Digital Economy Academy (IDEA)

## Abstract

The knowledge of protein interactions is crucial but challenging for drug discovery applications. This work focuses on protein interface prediction, which aims to determine whether a pair of residues from different proteins interact. Existing data-driven methods have made significant progress in effectively learning protein structures. Nevertheless, they overlook the conformational changes (i.e., flexibility) within proteins upon binding, leading to poor generalization ability. In this paper, we regard the protein flexibility as an *attack* on the trained model and aim to defend against it for improved generalization. To fulfill this purpose, we propose ATProt, an adversarial training framework for protein representations to robustly defend against the attack of protein flexibility. ATProt can theoretically guarantee protein representation stability under complicated protein flexibility. Experiments on various benchmarks demonstrate that ATProt consistently improves the performance for protein interface prediction. Moreover, our method demonstrates broad applicability, performing the best even when provided with testing structures from structure prediction models like ESMFold and AlphaFold2.

## 1 Introduction

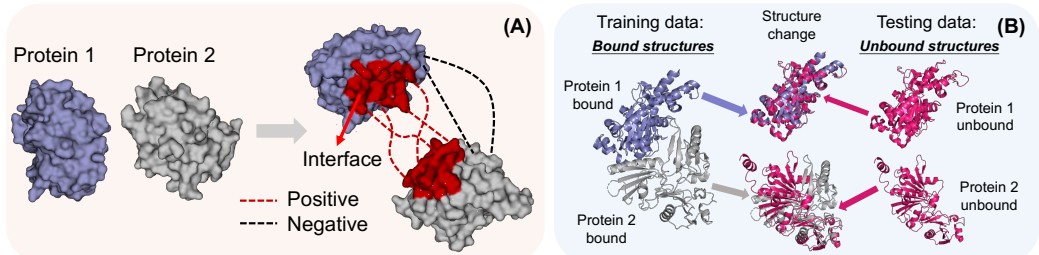

Figure 1: (A). **The task illustration.** PIP involves predicting if there is an interaction between two residues from different proteins. (B). **The task challenge.** During training, the input consists of bound structures of two proteins. However, for testing, one can only access their unbound structures.

Protein-protein interactions are important for understanding biological processes, and for the design of novel therapies [48, 16] and drugs [47]. The protein interface refers to the surface region of a protein where the interaction occurs. It therefore holds the key to revealing the specific interaction mechanism and understanding protein functions. In this work, we tackle the problem of protein

---

*Correspondence to: Zijing Liu (`liuzijing@idea.edu.cn`) and Jia Li (`jialee@ust.hk`).

38th Conference on Neural Information Processing Systems (NeurIPS 2024).

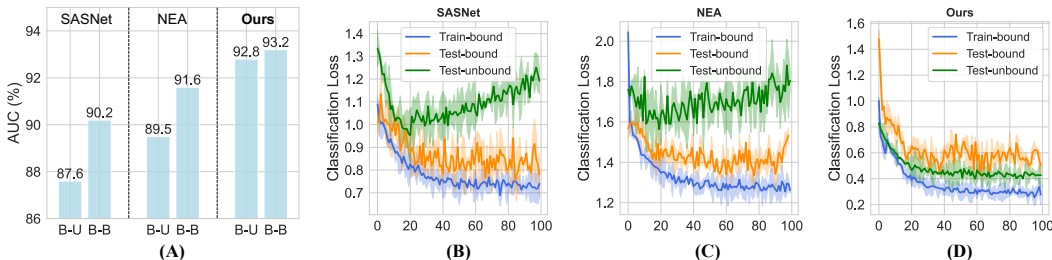

Figure 2: The impact of flexibility on results with the DB5.5 dataset [45]. (A) The testing results of two baselines (SASNet [43], NEA [15]) and our method. 'B-U' represents the popular formulation, i.e., training with bound structures and testing with unbound ones. 'B-B' refers to the formulation where both training and testing are conducted with bound structures. (B) Loss trends for three method.

interface prediction (PIP) (shown in Figure 1(A)): predicting whether two residues, each from an individual protein, interact with each other, given the separate structures of two proteins.

For years, data-driven methods based on deep learning (DL) have made significant progress in response to this critical task by effectively learning protein structures using geometric graph neural networks (GNN) [15, 33], 3D CNN [43], etc. Limited by the difficulty in accessing protein structure data, they typically follow a formulation of training on bound (after binding) structures and testing on unbound (before binding) ones, as depicted in Figure 1(B). For training, large-scale datasets like the Database of Interacting Protein Structures (DIPS) [43] typically consist of only bound structures, which are directly extracted from the PDB database[4]. In contrast, in the practical inference scenario, the model cannot access the bound structures but can only be provided with unbound ones [43, 15, 33]. In this paper, we empirically find that this training (bound)-testing (unbound) formulation leaves significant room for performance improvement. In Figure 2, exploratory experiments show that prevailing PIP methods are sensitive to flexibility. Utilizing the bound version structures for testing can greatly boost their performance. Based on these findings, we aim to answer the question in this paper—*how to handle the mismatch between bound and unbound structures for PIP?*

Since it is usually impractical to access the protein bound structures for testing, the most intuitive solution is to explicitly learn the mapping relationship from unbound to bound structures of proteins. However, this is challenging due to the following two factors: (1) The amount of pairwise unbound and bound structure data for proteins is extremely limited [10] (to our knowledge, only DB5.5 [45]). (2) A protein's bound structure is not unique and depends on its binding partner, so diverse training data is necessary. **To address this issue, we take a different route.** We consider any potentially complicated flexibility in a protein as an attack [29, 24], which can harm the testing performance of a model trained on bound structures. Therefore, our core idea is to enable protein representations with adversarial robustness, which can defend against the attacks of protein flexibility. In simple terms, for a protein with both unbound and bound versions, the model outputs similar (stable) representations.

In this work, we take an important step forward in mitigating the impact of flexibility on PIP. We propose **ATProt**, an end-to-end adversarial training (AT) framework for protein representations, to effectively defend against protein flexibility in PIP. Inspired by the recent protein graph representation methods [17, 53, 15], our model comprises graph-based feature extractors (encoders) for protein graphs. Our ATProt framework does not require computationally expensive data augmentation and can be smoothly applied to most existing protein graph encoders. Specifically, we implement differentiable AT regularizations for various protein representation encoders. Importantly, we introduce a novel and expressive graph encoder for protein representations and propose its theoretical regularization form for the first time. ATProt can produce stable representations for the same protein with different structure versions (e.g., bound, unbound, and model-generated ones). Extensive experiments on several protein interaction benchmarks verify that our ATProt method consistently outperforms advanced PIP methods. The results demonstrate the effectiveness of the AT regularization. Furthermore, ATProt maintains excellent performance even when tested with structures generated by AlphaFold2 [25] and ESMFold [32], allowing for user-friendly inference without the need for native structures.

## 2   Related Work

**Protein interface prediction.**   Protein interface prediction (PIP), a well-studied problem, focuses on determining whether there is an interaction between amino acids from two different proteins. Recently, a series of methods based on protein [8, 19, 17, 20, 14] or amino acid representation [39, 46] learning have achieved significant success.  NEA [15] pioneers the use of protein graphs to address PIP, where protein structure information is represented and aggregated, followed by the dense layers. SASNet [43] considers embedding the hierarchical structures of proteins, integrating atomic and amino acid information into a 4D-grid data, and employs 3D CNN for learning. To further enhance performance, more fine-grained structure information modeling, specifically surface geometry [46, 39], is introduced to effectively learn amino acid representations. Existing methods have effectively represented proteins from various perspectives of protein information. However, we have observed that protein flexibility, which is overlooked by most methods, poses significant performance bottlenecks for them. We focus on this key issue of mitigating the bound-unbound mismatch in protein structures to improve model generalization.

**Modelling protein flexibility.**   Recently, pioneer works in biology confirm that protein-protein interaction (PPI) conforms to the "induced fit" theory [27, 38]. Specifically, proteins undergo structure changes due to residue-level forces, and they adjust structures to achieve the best binding state. More importantly, proteins with PPI typically undergo larger structure changes at the interface compared to non-interface regions [10, 11, 52, 12], which will exacerbate the generalization challenge of the PIP task. Modelling flexibility directly is challenging, whether using traditional computational or deep learning (DL) approaches. Traditional methods often rely on finding the lowest energy state [42, 51] or introducing an induced fit model (specifically, the elastic network model) [12, 3] to guide structure deformations. The optimization space in these methods is vast, making them very time-consuming. DL-based methods [10] struggle to achieve satisfactory accuracy in learning the distribution mapping of bound-unbound states due to limited training data (i.e., 253 complexes in the DB5.5). As directly predicting bound structures is challenging, in the context of the PIP task, we choose to eliminate the influence of different versions of the same protein structure on the task.

## 3   Preliminaries

In this section, **(1)** we present the definition of the protein interface prediction (PIP). Then, **(2)** we verify the importance of representation stability through empirical and mathematical views. Finally, **(3)** we investigate to ensure protein representation stability within the adversarial training framework.

**Problem definition.**   We are given as input two proteins $\mathcal{P}^1$ and $\mathcal{P}^2$, consisting of $M$ and $N$ residues, respectively. The proteins are represented as their residue sequences and 3D structures, which are composed of $\alpha$-carbon atom locations of all residues. The goal of the PIP is to classify all possible pairs of residues from separate proteins. More formally, the set of data is $\{(\mathcal{P}^1_i, \mathcal{P}^2_j), y_{ij}\}_{1 \leq i \leq M, 1 \leq j \leq N}$, where $\mathcal{P}^1_i$ represents the $i$-th residue in protein $\mathcal{P}^1$ and $y_{ij} \in \{0, 1\}$.

### 3.1   The importance of protein representation stability

We verify the importance of stable protein representations from both empirical and theoretical perspectives. For clarity, we use the notations $\boldsymbol{X}^b_1, \boldsymbol{X}^b_2$ to represent the native bound structures of proteins $\mathcal{P}^1$ and $\mathcal{P}^2$, while $\boldsymbol{X}^t_1, \boldsymbol{X}^t_2$ represent their structures used for testing. After using a protein graph encoder, we have their representations, denoted as $\boldsymbol{H}^b_1, \boldsymbol{H}^b_2, \boldsymbol{H}^t_1, \boldsymbol{H}^t_2$. We denote the protein representation perturbation as $\|\delta_1\|_p + \|\delta_1\|_p$, where $\delta_1 = \boldsymbol{H}^t_1 - \boldsymbol{H}^b_1, \delta_2 = \boldsymbol{H}^t_2 - \boldsymbol{H}^b_2$.

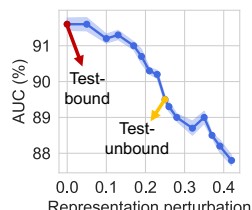

Figure 3: AUC vs. representation perturbation.

**From an empirical perspective,** in Figure 3, we quantify the test results of the NEA method [15] under flexibility. We gradually increase the structure change of test samples and calculate the representation perturbation. We note that the test performance is negatively correlated with the representation perturbation caused by flexibility. Moreover, testing with bound structures yields the best results.

**From a theoretical perspective,** we draw the consistent conclusion that stable representations lead to improved PIP results. Following [15], we model PIP as a pairwise classification problem. Specifically, we concatenate the $i$-th and $j$-th rows of $\boldsymbol{H}_1^t$ and $\boldsymbol{H}_2^t$ into a vector embedding, which is then sent to the PIP classifier $f$. In this case, the following proposition describes the impact of representation perturbations on PIP results.

**Proposition 3.1.** *For two proteins with $M$ and $N$ residues respectively, the classification results obtained using bound and unbound structures are the same. This is true if $N \left\| \delta_1 \right\|_p^p + M \left\| \delta_2 \right\|_p^p < \mathcal{A}(f, p)$, where $\mathcal{A}(f, p)$ can be a constant depending only on the PIP classifier $f$ and the norm $\left\| \cdot \right\|_p$.*

We detail $\mathcal{A}(f, p)$ and prove Proposition 3.1 in Appendix B.1. This proposition tells that stable protein representations (i.e., smaller $\left\| \delta_1 \right\|_p$ and $\left\| \delta_2 \right\|_p$) under flexibility are necessary for achieving high performance. Thus, to effectively address the structure mismatch in PIP, an intuitive idea is to perform adversarial training for protein representation learning.

## 3.2 Adversarial training

Here, we introduce the concept of adversarial training (AT) [1, 6, 18, 5, 7] and establish a connection between it and the stability of protein representations. We consider a classification task with a given dataset $\mathcal{D} = \{(x_i, y_i)\}_{i=1}^n$, consisting of $K$ classes. We assume that the entire prediction pipeline includes a representation model (e.g., encoder) and a classifier. The concept of adversarial training (AT) requires the entire pipeline to perform well not only on $\mathcal{D}$ but also on the worst-case distribution near $\mathcal{D}$, as determined by a specific distance metric. More concretely, the AT that we primarily focus on in this paper is the $\ell_p$-robustness. For a given $p$ value and a finite $\epsilon > 0$, AT aims to train a pipeline that can correctly classifies $(x + \delta, y)$ for any $\left\| \delta \right\|_p \leq \epsilon$, where $(x, y)$ belongs to $\mathcal{D}$.

Among all AT methods, Lipschitz neural networks belong to a common and effective category. Specifically, an encoder function is considered to have Lipschitz continuity if a slight perturbation to the input of the encoder does not significantly change its output.

Formally, the definition of Lipschitz continuity is given by:

**Definition 3.1.** *(Lipschitz continuity in adversarial training) An encoder function* ENC *is said to be $C$-Lipschitz continuous w.r.t. norm $\left\| \cdot \right\|$ if for any two versions of inputs $\boldsymbol{x}_1, \boldsymbol{x}_2$,*

$$\left\| \text{ENC}(\boldsymbol{x}_1) - \text{ENC}(\boldsymbol{x}_2) \right\| \leq C \left\| \boldsymbol{x}_1 - \boldsymbol{x}_2 \right\|. \tag{1}$$

Lipschitz continuity explains the requirements of AT for a general representation learning encoder. In the context of protein graph representation, this can be modified to become the definition below.

**Definition 3.2.** *(Lipschitz continuity for protein representations) A protein representation encoder $\Psi(\cdot)$ has $C$-Lipschitz continuity w.r.t. norm $\left\| \cdot \right\|$ if for any two versions of structure inputs $\boldsymbol{X}^t, \boldsymbol{X}^b$ and the invariant residue feature input $\boldsymbol{F}$,*

$$\left\| \Psi(\boldsymbol{F}, \boldsymbol{X}^t) - \Psi(\boldsymbol{F}, \boldsymbol{X}^b) \right\| \leq C \left\| \boldsymbol{X}^t - \boldsymbol{X}^b \right\|. \tag{2}$$

## 3.3 How to ensure Lipschitz continuity for protein graph representations?

As an expressive representation, graph structured data is widely used for representing input proteins [17, 22, 19], with residues acting as nodes and physical interactions as edges. Let $\mathcal{G} = (\mathcal{V}, \mathcal{E})$ be an undirected graph with nodes $\mathcal{V} = \{1, ..., N\}$, edges $\mathcal{E} \subset \mathcal{V}^2$, graph signal $\boldsymbol{F} \in \mathbb{R}^{N \times d}$ and a graph shift operator $\boldsymbol{L} \in \mathbb{R}^{N \times N}$ (i.e., node connectivity). We consider any variant of the spectral GNN (e.g., GCN [26, 30], ChebNets [13], BWGNN [41, 40]) that follows the concept of learning filter coefficients for graph convolution. By constructing the filter $h(\boldsymbol{L}) := \sum_{k=0}^K \theta_k \boldsymbol{L}^k$ ($\theta_k$ are learnable parameters), the protein graph representation can be defined as $\boldsymbol{H} = \sum_{k=0}^K \theta_k \boldsymbol{L}^k \boldsymbol{F} := h(\boldsymbol{L})\boldsymbol{F}$.

Here, we extend the Definition 3.2 to the scenario of using GNN models. To achieve this, we assume $\boldsymbol{L}$ is perturbed to become $\tilde{\boldsymbol{L}}$ due to the protein flexibility, and introduce the key factor (***GNN filter stability constant*** $C_h$), for achieving GNN-based stable protein representations.

**Definition 3.3.** *(GNN filter stability constant) Given a graph spectral filter $h : [0, 2] \mapsto [0, 1]$, it is defined as Lipschitz with constant $C_h > 0$ if for any pair of points $\lambda_1, \lambda_2$:*

$$\left| h(\lambda_1) - h(\lambda_2) \right| \leq C_h \left| \lambda_1 - \lambda_2 \right|, \tag{3}$$

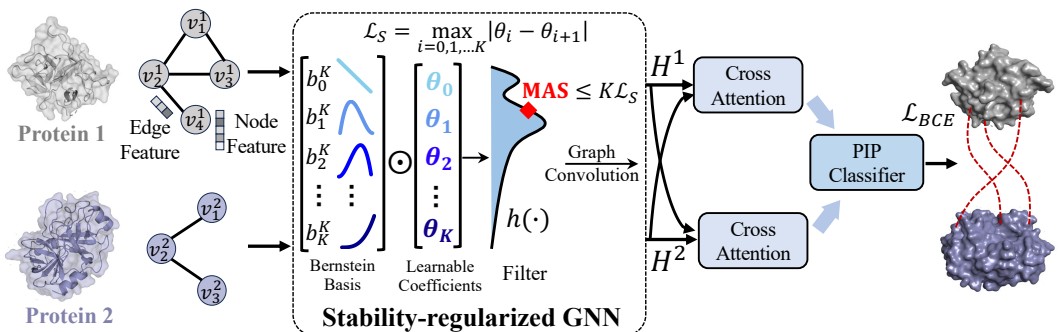

Figure 4: **The framework overview with the BernNet encoder.** The whole framework contains the stability-regularized graph encoder for stable protein representations, the cross attention layers for communication and the final binary classifier. ATProt takes in two protein graphs as inputs, and extracts features with the pre-defined graph encoder (BernNet is taken as an example here). The PIP results are obtained after the learned representations have passed through the cross attention module and classifier. The $\mathcal{L}_S$ loss for stability regularization and classification loss $\mathcal{L}_{BCE}$ jointly optimize the model.

which introduces our main theorem below.

**Theorem 3.1.** *(Protein graph stability with GNNs) Let the perturbation to $\boldsymbol{L}$ is finite such that $\left\| \tilde{\boldsymbol{L}} - \boldsymbol{L} \right\|_p \leq \epsilon$. The protein graph encoder $\Psi(\cdot)$ is always stable with a polynomial filter $h$ if for some finite $C_h$,*

$$\left\| \Psi(\boldsymbol{F}, \boldsymbol{L}) - \Psi(\boldsymbol{F}, \tilde{\boldsymbol{L}}) \right\|_p \leq \epsilon C_h \cdot \mathcal{A}(\Psi) \cdot \|\boldsymbol{F}\|_p, \tag{4}$$

*where $\mathcal{A}(\Psi)$ is a constant determined by the model (e.g., layer number and feature dimension).*

Based on Eq. 3 and Eq. 4, the lower bound of the left term in Eq. 4 is solely determined by the maximum of $C_h$ (denoted as $C_h^*$). To be more intuitive, $C_h^*$ is equal to the maximum absolute slope (MAS) of the filter $h$ (the straightforward proof is in the Appendix B.2).

Therefore, we conclude the core idea for designing our ATProt framework as follows:

> *We can enhance the stability of protein representations by decreasing the MAS value of $h$.*

## 4 Method

**Overview.** We propose ATProt, an end-to-end framework (illustrated in Fig. 4) to boost PIP from a view of representation stability. Specifically, our model inputs two proteins whose structures can be provided in various sources (e.g., native bound, native unbound, AlphaFold2, ESMFold). ATProt incorporates a protein graph representation encoder and a targeted differentiable regularization scheme to theoretically guarantee the representation stability. Our ATProt framework can be implemented with at least four protein graph encoders, and we provide specific examples of these cases. Lastly, for the PIP task, we apply a cross-attention module to facilitate communication between the protein representations and use a simple linear classifier for final prediction.

**Protein representation.** We represent a protein as an undirected weighted graph $\mathcal{G} = (\mathcal{V}, \mathcal{E})$. Each node $v_i \in \mathcal{V}$ representing one residue has a $d$-dimensional feature vector $\boldsymbol{F}_i \in \mathbb{R}^d$ (i.e., residue type) and a 3D coordinate $\boldsymbol{X}_i \in \mathbb{R}^3$ (i.e., the $\alpha$-carbon atom location). Edges $\mathcal{E} = \{(i, j)\}$ are constructed with a k-nearest-neighbor (k-NN) graph using Euclidean distance. To distinguish the edges, we follow [17] to construct the SE(3)-invariant edge features $\{e_{i,j} : \forall (i, j) \in \mathcal{E}\}$.

**The goal of ATProt.** According to Definition 3.2, we aim to achieve representation stability, which means that the perturbation in the representation caused by protein flexibility is constrained. In

addition, it is also important to ensure the commonly encapsulated SE(3)-invariance, which means that the representation of the protein is not affected by its rotation or translation. Formally, we represent a single protein $\mathcal{P}$ consisting of $N$ residues, with its residue-level feature matrix $\boldsymbol{F} \in \mathbb{R}^{N \times d}$ and residue-level structure $\boldsymbol{X} \in \mathbb{R}^{N \times 3}$. We wish our model $\Psi(\cdot)$ to ensure the following property.

$$
\begin{aligned}
&\text{Given} \quad \boldsymbol{Z}, \boldsymbol{H} = \Psi(\boldsymbol{F}, \boldsymbol{X}); \boldsymbol{Z}', \boldsymbol{H}' = \Psi(\boldsymbol{F}, \boldsymbol{Q}\boldsymbol{X} + \boldsymbol{g} + \Delta\boldsymbol{X}), \\
&\text{we have} \quad \left\| \boldsymbol{H} - \boldsymbol{H}' \right\| \leq C \cdot \left\| \Delta\boldsymbol{X} \right\|, (C\text{-Lipschitz continuity}) \\
&\forall \boldsymbol{Q} \in SO(3), \forall \boldsymbol{g} \in \mathbb{R}^3, \forall \Delta\boldsymbol{X} \in \mathbb{R}^{3 \times N}, \exists C \in \mathbb{R}.
\end{aligned}
\tag{5}
$$

## 4.1 Adversarial training regularizations for various encoders

The Fourier transform is a powerful tool in the representation of both structured [50, 26, 49, 28] and unstructured data [9, 21, 31]. Our main focus is on employing spectral graph encoders for representing protein graphs. These encoders adhere to the principle of utilizing a graph spectral filter $h(\lambda) = \sum_{k=0}^{K} \theta_k b_k(\lambda)$ to learn effective protein representations. Here, $\{b_k\}_{k=0}^{K}$ is the pre-defined filter basis and $\{\theta_k\}_{k=0}^{K}$ is the learnable parameters. For graph convolution, the filter $h(\cdot)$ will be applied to the whole Laplacian matrix $\boldsymbol{L}$, which is calculated from the protein structure data. Specifically, to construct $\boldsymbol{L}$, we first apply a Multi-Layer Perceptron (MLP) to reduce the dimensionality of $\boldsymbol{E}$ to 1, and incorporate the results into the edge weight matrix $\boldsymbol{W}$:

$$
\boldsymbol{W}_{i,j} = \begin{cases} \text{MLP}_e(e_{i,j}), & (i,j) \in \mathcal{E} \\ 0, & (i,j) \notin \mathcal{E} \end{cases}
\tag{6}
$$

Then the Laplacian matrix $\boldsymbol{L}$ is obtained by $\boldsymbol{L} = \boldsymbol{I} - \boldsymbol{D}^{-1/2}\boldsymbol{W}\boldsymbol{D}^{-1/2}$, where $\boldsymbol{D}$ is the degree matrix, i.e., $\boldsymbol{D} = \text{diag}(\sum_j \boldsymbol{W}_{1,j}, ..., \sum_j \boldsymbol{W}_{N,j})$.

The protein graph representation can be defined as:

$$
\boldsymbol{H} = \sum_{k=0}^{K} \theta_k b_k(\boldsymbol{L})\boldsymbol{F} = h(\boldsymbol{L})\boldsymbol{F},
\tag{7}
$$

where the result of $h(\boldsymbol{L})$ represents the graph spectral response.

In this paper, we introduce four types of top graph encoders (i.e., Simple GCN [49], Chebynet [13], Low-pass filter [34], and BernNet [23]) along with their corresponding stability regularizations.

**Case 4.1.** *(Simple GCN encoder [49]) The Simple GCN encoder (SGC) utilizes a spectral filter in the monomial function form:*

$$
h(\lambda) = \lambda^K.
\tag{8}
$$

*Since the spectrum $\lambda$ lies in $[-1, 1]$, it is clear that the maximum absolute slope (MAS) that $h(\lambda)$ can reach is $K$. Therefore, the stability regularization of the SGC encoder does not involve any loss function. We can directly constrain the size of the order $K$.*

**Case 4.2.** *(Chebynet encoder [13]) The Chebynet encoder utilizes a spectral filter in the polynomial function form:*

$$
h(\lambda) = \sum_{k=0}^{K} \theta_k \lambda^k.
\tag{9}
$$

*The stability regularization of Chebynet can be implemented by the loss function $\mathcal{L}_S = \sum_{k=1}^{K} k|\lambda^k|$.*

**Case 4.3.** *(Low-pass filter encoder [34]) The Low-pass filter (LPF) encoder utilizes a spectral filter in the low-pass filter function form:*

$$
h(\lambda) = (1 + \theta\lambda)^{-1}.
\tag{10}
$$

*The stability regularization of LPF can be implemented by the loss function $\mathcal{L}_S = \theta$.*

**Case 4.4.** *(BernNet encoder [23]) The BernNet encoder utilizes Bernstein basis, the state-of-the-art graph spectral basis, to construct the graph spectral filter:*

$$
h(\lambda) = \sum_{k=0}^{K} \theta_k b_k^K(\lambda^{(l+1)}) = \sum_{k=0}^{K} \theta_k \frac{1}{2^K} \binom{K}{k} (2\boldsymbol{I} - \lambda^{(l+1)})^{K-k}(\lambda^{(l+1)})^k.
\tag{11}
$$

*According to Theorem 3.1, the BernNet encoder easily satisfies the representation C-Lipstchiz continuity in Eq. 4 provided that the filter $h(\lambda)$ always has a finite MAS. However, in Eq. 11, the analytical relationship between MAS and $\{\theta_k\}_{k=0}^K$ is intractable, suggesting the difficulty of constraining $C$ with a gradient descent manner. Thus, our next goal is to discover a differentiable method for constraining the minimum of $C$ in the training process.*

The stability of the BernNet is still unclear in these four cases, prompting us to investigate its regularization form. To the best of our knowledge, it is the first investigation of the Lipstchiz continuity for Bernstein-based spectral filters.

## 4.2 Guaranteeing stability of the BernNet encoder

For clarity, we rewrite the Eq. 4 as $\left\| \boldsymbol{H} - \boldsymbol{H}' \right\| \leq C \cdot \|\Delta\boldsymbol{X}\| = C_h \cdot \mathcal{A}(\Psi) \cdot \|\Delta\boldsymbol{X}\| \cdot \|\boldsymbol{F}\|$, where, referring to Eq. 3, $\mathcal{A}(\Psi)$ is determined by the model architecture hyperparameters and remains constant. We say the overall model $\Psi$ is of $C$-Lipstchiz continuity and the learned spectral filter $h(\lambda) = \sum_{k=0}^K \theta_k b_k^K(\lambda)$ is of $C_h$-Lipstchiz continuity. We aim to constrain the minimum of constant $C_h$ (denoted as $C_h^*$) with $\{\theta_k\}_{k=0}^K$ by discovering the underlying relationship between them. Finally, we propose an auxiliary differentiable regularization of $\{\theta_k\}_{k=0}^K$ to constrain $C_h^*$ to a controllable bound, for any $\Delta\boldsymbol{X}$.

**Theorem 4.1.** *Given an arbitrary polynomial function $f(\lambda)$ on $\lambda \in [0, 2]$ and suppose its $K$-order Bernstein polynomial is denoted as $h(\lambda) = \sum_{k=0}^K f(\frac{2k}{K}) \binom{K}{k} (2-\lambda)^{K-k}\lambda^k$. For any point pair $\lambda_1, \lambda_2 \in [0, 2]$, if there exists a constant $C_f$ for $|f(\lambda_1) - f(\lambda_2)| \leq C_f|\lambda_1 - \lambda_2|$, then $h(\lambda)$ always holds $C_f$-Lipstchiz continuity for all $K \geq 1$:*

$$|h(\lambda_1) - h(\lambda_2)| \leq C_f |\lambda_1 - \lambda_2|. \tag{12}$$

We introduce Theorem 4.1, which describes the stability relationship between the filter $h(\lambda)$ and an auxiliary function $f(\lambda)$. It tells that with Berstein basis, the MSA of $h(\lambda)$ will never exceed that of the auxiliary function $f(\lambda)$. Due to $f(\frac{2k}{K}) = \theta_k$, for all $k \in [0, K] \cap \mathbb{Z}$, $f(\lambda)$ can be any 2-D curve passing through all points of $\{(\frac{2k}{K}, \theta_k)\}_{k=0}^K$. Therefore, the MAS of $f(\lambda)$ is analytically tractable, and we can subsequently provide the bounds for the MAS of $h(\lambda)$. We have the following proposition, which is accompanied by a detailed derivation in Appendix B.3.

**Proposition 4.1.** *Suppose $h(\lambda)$ is approximated with Bernstein basis, i.e., $h(\lambda) = \sum_{k=0}^K \theta_k b_k^K(\lambda)$. Denoting the MAS (minimum Lipschitz constant) of $h$ as $C_h^*$, it can be upper bounded by*

$$C_h^* \leq \max_{i \in [0, K-1] \cap \mathbb{Z}} K \cdot |\theta_i - \theta_{i+1}|. \tag{13}$$

To summarize, Bernstein basis applied to our ATProt guarantees a finite MAS, and more importantly, we can further enhance the stability of the BernNet encoder with the following objective:

$$\mathcal{L}_\mathcal{S} = \max_{i \in [0, K-1] \cap \mathbb{Z}} |\theta_i - \theta_{i+1}|. \tag{14}$$

## 4.3 Protein interface prediction

Given proteins $\mathcal{P}_1, \mathcal{P}_2$ with their initial feature $\boldsymbol{F}_1, \boldsymbol{F}_2$ and coordinates $\boldsymbol{X}_1, \boldsymbol{X}_2$, ATProt produces their stable representations under respective structure perturbations.

$$\boldsymbol{H}_1 \in \mathbb{R}^{M \times d} = ATProt(\boldsymbol{X}_1, \boldsymbol{F}_1); \boldsymbol{H}_2 \in \mathbb{R}^{M \times d} = ATProt(\boldsymbol{X}_2, \boldsymbol{F}_2). \tag{15}$$

We apply a cross-attention layer (shown in Appendix D) to enable communication between proteins and obtain their final representations $\boldsymbol{H}_1', \boldsymbol{H}_2'$. Next, we aim to predict whether pairs of inter-protein residues belong to the interface, which involves performing pairwise binary classification. Concretely, for each training pair of proteins, we have a set of $10N_I$ labeled pairs $\left\{(([\boldsymbol{H}_1']_i, [\boldsymbol{H}_2']_i), y_i)\right\}_{i=1}^{10N_I}$, where $y_i \in \{0, 1\}$, $N^I$ is the number of positive residue pairs and $9N^I$ negative ones are downsampled. We take the element-wise product of two residue representations and feed it to another MLP with the Sigmoid function to compute the probability $p_i$. Weighted cross-entropy loss is used for training.

$$\mathcal{L}_I = \frac{1}{|\mathcal{Y}_{train}|} \sum_{y^k \in \mathcal{Y}_{train}} \left( \sum_{i=0}^{2N_I^k} -y_i^k \log p_i^k - (1 - y_i^k)\log(1 - p_i^k) \right). \tag{16}$$

# 5 Experiments

## 5.1 Experimental setup

**Datasets and processing.** We evaluate our method on the complexes from Docking Benchmark 5.5 (DB5.5) [45], a gold standard dataset with high-quality, and Database of Interacting Protein Structures (DIPS) [43], which collects 41,876 complexes mined from PDB [4]. DB5.5 only contains 253 complex structures manually curated by domain experts, which cover both native unbound and bound structures. In comparison, DIPS has a significantly larger data size, but it only includes bound structures of proteins. The two datasets are randomly divided into training, validation, and testing sets with the following sizes: 203/25/25 (DB5.5) and 39,937/974/965 (DIPS).

For both datasets, we test using various versions of protein structures as inputs for PIP. To accomplish this, we prepared three versions of the testing set for DB5.5, including native unbound structures, structures produced by AlphaFold2, and structures produced by ESMFold. We use `Native-Bound`, `Native-Unbound`, `ESMFold`, `AlphaFold2` to denote these four settings respectively.

As for DIPS, since it does not have native unbound structures, we only used ESMFold to prepare unbound structure inputs for its testing set. AlphaFold2 is not considered due to its high computational cost for the entire testing set of DIPS.

**Baselines.** We compare our ATProt method with state-of-the-art conventional machine learning method BIPSPI [36], the CNN-based methods Siamese Atomic Surfacelet Network (SASNet) [43], Diffusion-Convolutional Neural Networks (DCNN) [2], differentiable molecular surface interaction fingerprinting (dMaSIF) [39], and a set of GNN-based methods Deep Tensor Neural Networks (DTNN) [37], and NEA [15].

**Setup and metrics.** We consider three experimental setups: (1) performing training and testing both on the DB5.5; (2) performing training and testing both on the DIPS; and (3) performing pre-training on the DIPS, fine-tuning on the DB5.5 and testing on the DB5.5. For our proposed ATProt method, we consider graph encoders of Simple GCN, Chebynet, Low-pass filter and BernNet (denoted as ATProt-SGC, ATProt-Cheby, ATProt-LPF, and ATProt-Bern, respectively).

For each complex in the testing set, assuming that the two proteins have $M$ and $N$ residues, respectively, we test all its $M \times N$ binary classification samples and calculate the Area Under the ROC Curve (AUC) value. Following [15], we report the median AUC score (MedAUC) across all complexes as the final evaluation metric.

Table 1: **Training and testing on the DB5.5.** Mean and standard deviation values of the MedAUC scores of all baselines, computed from three random seeds. The best performance is in **bold** and the second best one is underlined. 'SR' means the proposed stable regularization $\mathcal{L}_S$.

| Methods | Native-Bound | Native-Unbound | ESMFold | AlphaFold2 |
|---|---|---|---|---|
| BIPSPI [36] | **0.937 (0.008)** | 0.911 (0.017) | 0.896 (0.008) | 0.887 (0.013) |
| SASNet [43] | 0.902 (0.007) | 0.876 (0.017) | 0.887 (0.025) | 0.881 (0.020) |
| dMaSIF [39] | 0.928 (0.005) | 0.912 (0.009) | 0.906 (0.003) | 0.892 (0.012) |
| DTNN [37] | 0.912 (0.005) | 0.886 (0.007) | 0.883 (0.010) | 0.878 (0.021) |
| NEA [15] | 0.916 (0.015) | 0.895 (0.009) | 0.902 (0.010) | 0.883 (0.012) |
| **ATProt-SGC** | 0.925 (0.015) | 0.918 (0.004) | 0.909 (0.012) | 0.924 (0.014) |
| **ATProt-Cheby** | 0.928 (0.017) | 0.922 (0.007) | 0.922 (0.005) | 0.924 (0.011) |
| **ATProt-LPF** | 0.915 (0.017) | 0.919 (0.019) | 0.911 (0.009) | 0.911 (0.010) |
| **ATProt-Bern** | 0.932 (0.017) | **0.928 (0.014)** | **0.929 (0.014)** | **0.925 (0.011)** |
| ATProt-Bern w/o SR | 0.934 (0.009) | 0.901 (0.011) | 0.897 (0.010) | 0.901 (0.012) |

**Results.** Table 1, 2, and Table 3 (shown in Appendix) show the model performance for PIP. We find that our method is competitive and outperforms the majority of baseline methods with native bound structures. Under this `Native-Bound` setting, although ATProt is slightly less effective than BIPSPI, it demonstrates the ability to learn sufficiently powerful protein representations (especially with BernNet and Chebynet). Notably, all the baselines fail significantly when using non-bound structures

Table 2: **Pre-training on the DIPS, fine-tuning on the DB5.5 and finally testing on the DB5.5.**
▲/▼ indicates that the model performs better/worse than without pre-training (i.e., results in Table 1).

| Methods | Native-Bound | Native-Unbound | ESMFold | AlphaFold2 |
|---|---|---|---|---|
| BIPSPI [36] | **0.945 (0.017)**▲ | 0.913 (0.009)▲ | 0.907 (0.007)▲ | 0.903 (0.010)▲ |
| SASNet [43] | 0.916 (0.003)▲ | 0.899 (0.011)▲ | 0.900 (0.016)▲ | 0.878 (0.012)▼ |
| dMaSIF [39] | 0.922 (0.006)▼ | 0.903 (0.012)▼ | 0.901 (0.006)▼ | 0.910 (0.007)▲ |
| DTNN [37] | 0.919 (0.003)▲ | 0.893 (0.004)▲ | 0.896 (0.013)▲ | 0.898 (0.015)▲ |
| NEA [15] | 0.926 (0.013)▲ | 0.911 (0.009)▲ | 0.897 (0.014)▼ | 0.892 (0.014)▲ |
| ATProt w/o SR | 0.939 (0.013)▲ | 0.910 (0.016)▲ | 0.902 (0.011)▲ | 0.904 (0.014)▲ |
| **ATProt-Bern** | 0.939 (0.011)▲ | **0.934 (0.014)**▲ | **0.937 (0.012)**▲ | **0.928 (0.012)**▲ |

as inputs, whereas our method maintains similar performance compared to that under `Native-Bound`. As can be seen from the results of 'ATProt-BernNet w/o SR', the robustness to structure perturbation is attributed to the proposed SR (stable regularization) strategy. Overall, the experiments on DB5.5 demonstrate that our method does not rely on native bound or unbound structure data for inference, but can directly utilize structure prediction software to achieve satisfactory PIP results. From Table 2, 3, we note that although DIPS has a larger data scale than DB5.5, it is difficult to achieve better results. However, pre-training on DIPS can improve most of the testing results on DB5.5.

In summary, all three tables show consistent performance degradation for all baselines with the non-bound structures. In contrast, our method can defend against this bound-unbound mismatch and achieve performance similar to, or even better than that with `Native-Bound`.

**Benefits of representation stability.** Figure 5 shows the dimensionality reduction visualization of residue representations with t-SNE [44]. We observe that ATProt leads to a "clustering" effect in the representations, which is the result of stability regularization. Importantly, this ultimately generates clearer classification boundary in visualization and quantification (i.e., the Silhouette Score [35]).

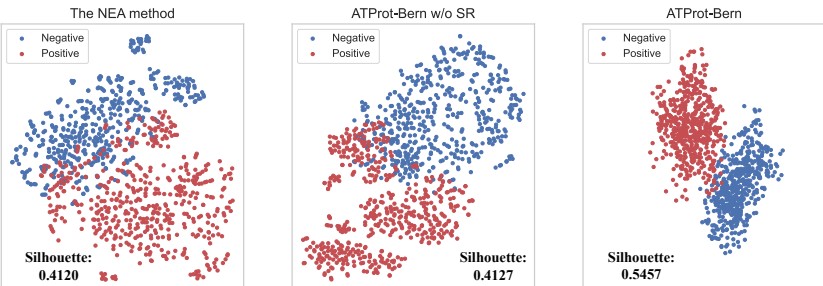

Figure 5: The t-SNE visualization for the last layer representations. The *x* and *y* axes of all three subplots are uniformly scaled to (0, 1).

## 6   Conclusion

In this paper, we present ATProt, an end-to-end learning framework for protein interface prediction (PIP). By highlighting the importance of protein representation stability for the PIP task, we tailor the stability regularization for four types of spectral graph encoders, theoretically ensuring that the ATProt framework exhibits Lipschitz continuity properties. Our method demonstrates competitive empirical performance compared to leading deep learning-based baselines, especially when dealing with significant bound-unbound structure gaps. Importantly, ATProt demonstrates the ability to perform inference in native-structure free scenarios.

**Limitations.** To present a clear and focused issue, we do not incorporate adversarial training-based classifiers into our framework. However, it is expected that their inclusion could further enhance the performance of ATProt, as they have shown promising results in the field of computer vision.

## Acknowledgement

This work was supported by HKUST-HKUST(GZ) 20 for 20 Crosscampus Collaborative Research Scheme C019, and Shenzhen Hetao Shenzhen-Hong Kong Science and Technology Innovation Cooperation Zone under Grant No. HTHZQSWS-KCCYB-2023052.

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

# Supplementary of "Towards Stable Representations for Protein Interface Prediction"

## A Representing proteins with graph data

Let $\mathcal{G} = (\mathcal{V}, \mathcal{E})$ be an undirected graph with nodes $\mathcal{V} = \{1, ..., N\}$, edges $\mathcal{E} \subset \mathcal{V}^2$, graph signal $\boldsymbol{F} \in \mathbb{R}^{N \times d}$ and a graph shift operator $\boldsymbol{L} \in \mathbb{R}^{N \times N}$ (i.e., node connectivity). We construct edges with the k-nearest neighbor (k-NN) algorithm and each node in $\mathcal{G}$ is connected to the 10 closest nodes within a physical distance of less than 30 Å. The edge attributes are distances between $\alpha$-carbon atoms encoded with Gaussian basis functions. The nodes have two kinds of attributes: the one-hot encoding of amino acid type and the surface-aware features at the residue level. The latter is defined by [17] to distinguish residues closer to the protein surface from those in the interior. Notably, we do not introduce any biochemically related attributes of atoms or amino acids featurization. Instead, protein graphs are constructed solely using their $\alpha$-carbon coordinates.

## B Proofs of main propositions

### B.1 Proof of Proposition 3.1

Assuming we model the protein interface prediction as a pairwise classification problem. Basically, this proposition provides a perturbation radius, within which if the perturbations of two protein representations fall, the predicted correspondences of all pairs of inter-protein residues will be robust.

Formally, we apply the model $\Psi$ to obtain $d$-dimensional protein representations $\boldsymbol{H}_1 = \Psi(\mathcal{P}_1) \in \mathbb{R}^{M \times d}$, $\boldsymbol{H}_2 = \Psi(\mathcal{P}_2) \in \mathbb{R}^{N \times d}$. The $i$-th row of $\boldsymbol{H}_1$ and $j$-th row of $\boldsymbol{H}_2$ are concatenated into $\boldsymbol{x}^{(ij)} \in \mathbb{R}^{2d}$, which is the pairwise residue representations. We then use a neural network $f$ to calculate the probability of the presence of residue correspondence, and define the classifier as $g(\boldsymbol{x}^{(ij)}) := \arg \max_k f_k(\boldsymbol{x}^{(ij)})$. Assuming that the structure changes of two proteins cause perturbations $\delta_1, \delta_2$ on $\boldsymbol{H}_1$ and $\boldsymbol{H}_2$, respectively, the interface prediction robustness can be determined by the following proposition. First, we rewrite Proposition 3.1 into a more formal expression.

**Proposition B.1.** *For all $i \in [1, M] \cap \mathbb{Z}$ and $j \in [1, N] \cap \mathbb{Z}$, the interface classifier $g$ is provably robust for arbitrary $\boldsymbol{x}^{(ij)}$ (i.e., $g$ is robust for all residue pairs of $\mathcal{P}_1$ and $\mathcal{P}_2$) if $N \|\delta_1\|_p^p + M \|\delta_2\|_p^p < \mathcal{A}(f, p)$, where $\mathcal{A}(f, p)$ can be a constant depending only on $f$ and the norm $\|\cdot\|_p$.*

*Proof.* . We denote $\boldsymbol{x}$ as a variable for convenience to represent the pairwise residue representation. Let $g(\boldsymbol{x}) = y$. Suppose there exists a perturbation $\boldsymbol{\delta}$ such that $g(\boldsymbol{x} + \boldsymbol{\delta}) \neq g(\boldsymbol{x})$ and $f_j(\boldsymbol{x} + \boldsymbol{\delta}) \geq f_y(\boldsymbol{x} + \boldsymbol{\delta})$ for some $j \neq y$. We first prove that $\|\delta\|_p \geq \frac{\sqrt[p]{2}}{2C} \cdot \text{margin}(f(\boldsymbol{x}))$, where $C$ is the lipschitz constant of $f$, $\text{margin}(f(\boldsymbol{x}))$ is the margin between the largest and second runner-up output logits. Define $\boldsymbol{z}' = f(\boldsymbol{x} + \boldsymbol{\delta})$, then $z'_y \leq z'_j$. The difference between outputs $\boldsymbol{z}$ and $f(\boldsymbol{x})$ can be lower bounded by

$$\|\boldsymbol{z}' - f(\boldsymbol{x})\|_p \geq \left\| (z'_y, z'_j)^T - ([f_y(\boldsymbol{x}), f_j(\boldsymbol{x})])^T \right\| = (|z'_y - f_y(\boldsymbol{x})|^p + |z'_j - f_j(\boldsymbol{x})|^p)^{\frac{1}{p}} \quad (17)$$

In the above equation, we utilize the fact that setting certain elements of a vector to zero can only decrease its $p$-norm. Let us now consider the following optimization problem:

$$\min_{\boldsymbol{z}'} |z'_y - f_y(\boldsymbol{x})|^p + |z'_j - f_j(\boldsymbol{x})|^p \qquad \text{s.t. } z'_y \leq z'_j \quad (18)$$

When $z'_y = z'_j = (f_y(\boldsymbol{x}) + f_j(\boldsymbol{x}))/2$, we have the minimum of (2) and the update for (11):

$$\|\boldsymbol{z}' - f(\boldsymbol{x})\|_p \geq \left\| (z'_y, z'_j)^T - ([f_y(\boldsymbol{x}), f_j(\boldsymbol{x})])^T \right\| \geq \frac{\sqrt[p]{2}}{2}(f_y(\boldsymbol{x}) - f_j(\boldsymbol{x})) \quad (19)$$

According to the Lipschitz constant of $f$, we have

$$\left\| \boldsymbol{z}' - f(\boldsymbol{x}) \right\|_p \leq C \left\| \delta \right\|_p \tag{20}$$

Considering (3) and (4), we have the initial conclusion as follows

$$\|\delta\|_p^p \geq \left( \frac{\sqrt[p]{2}}{2C} \cdot \mathrm{margin}(f(\boldsymbol{x})) \right)^p \tag{21}$$

Then we consider the complete residue pair representation set $\{\boldsymbol{x}^{(ij)}\}_{i \in [1,M] \cap \mathbb{Z}, j \in [1,N] \cap \mathbb{Z}}$.

$$\sum_{i=1}^{M} \sum_{j=1}^{N} \left\| \delta^{(ij)} \right\|_p^p \geq \sum_{i=1}^{M} \sum_{j=1}^{N} \left( \frac{\sqrt[p]{2}}{2C} \cdot \mathrm{margin}(f(\boldsymbol{x}^{(ij)})) \right)^p \tag{22}$$

We denote the perturbations on protein representations $\boldsymbol{H}_1$ and $\boldsymbol{H}_2$ as $\delta_1$ and $\delta_2$, respectively. $\delta^{(ij)}$ is actually the concatenation of the $i$-th row of $\delta_1$ and the $i$-th row of $\delta_2$, i.e., $\delta^{(ij)} = \mathrm{AGG}(\delta_1, \delta_2)$. Thus the left term of (6) is equal to $N \|\delta_1\|_p^p + M \|\delta_2\|_p^p$. Finally, we conclude the proof:

$$N \|\delta_1\|_p^p + M \|\delta_2\|_p^p \geq \sum_{i=1}^{M} \sum_{j=1}^{N} \left( \frac{\sqrt[p]{2}}{2C} \cdot \mathrm{margin}(f(\boldsymbol{x}^{(ij)})) \right)^p = MN \left( \frac{\sqrt[p]{2}}{2C} \cdot \mathrm{margin}^*(f(\boldsymbol{x})) \right)^p \tag{23}$$

where $\mathrm{margin}^*(f(\boldsymbol{x}))$ is the average value of all margins w.r.t the $MN$ concatenated residue representations. $\square$

### B.2  Proof of the statement after Theorem 3.1 that $C_h^*$ is equal to the MAS.

*Proof.* We prove this based on the definition of graph Lipschitz filter, which is continuously differentiable due to the polynomial approximation.

Let $C_h$ be the Lipschitz constant of $h$, and let $u(\lambda) = |h'(\lambda)|$. We want to show that $C_h$ is minimized when $C_h = \sup_\lambda u(\lambda)$.

Suppose there exist $\lambda_1$ and $\lambda_2$ such that $|\lambda_1 - \lambda_2| > 0$ and $|h(\lambda_1) - h(\lambda_2)| > C_h |\lambda_1 - \lambda_2|$. Then, by the mean value theorem, there exists $\lambda_3$ between $\lambda_1$ and $\lambda_2$ such that $|h'(\lambda_3)| > C_h$. But this contradicts the assumption that $C_h$ is the Lipschitz constant of $h$. Therefore, $C_h$ must be greater than or equal to $\sup_\lambda u(\lambda)$.

To show that $C_h$ is minimized when $C_h = \sup_\lambda u(\lambda)$, suppose there exists a Lipschitz constant $C_h'$ such that $C_h' < \sup_\lambda u(\lambda)$. Then, for any $\lambda_1$ and $\lambda_2$, we have

$$\begin{aligned}
|h(\lambda_1) - h(\lambda_2)| &\leq C_h' |\lambda_1 - \lambda_2| \\
&< \sup_\lambda u(\lambda) |\lambda_1 - \lambda_2| \\
&\leq |h'(\lambda_3)| |\lambda_1 - \lambda_2|
\end{aligned} \tag{24}$$

where $\lambda_3$ is some point between $\lambda_1$ and $\lambda_2$. But this contradicts the definition of $u(\lambda)$ as the maximum of $|h'(\lambda)|$. Therefore, $C_h$ must be equal to $\sup_\lambda u(\lambda)$.

Thus, we have shown that $C_h$ is minimized when $C_h = \sup_\lambda u(\lambda)$, as desired. For readability, we denote the minimum of $C_h$ as $C_h^*$ and $\sup_\lambda u(\lambda)$ as the MAS (maximum absolute slope) of $h$, respectively. $\square$

### B.3  Proof of Proposition 4.1

We start the analysis with the unique stability-preservation property of the Bernstein basis shown in Theorem 4.1. Simply put, during Bernstein polynomial approximation, the outcome polynomial $h(\lambda)$ is at least as stable as the target function $f(\lambda)$. This is crucial as it tells that the MAS of an exact function $f(\lambda)$ can always serve as an upper bound for $C_h^*$.

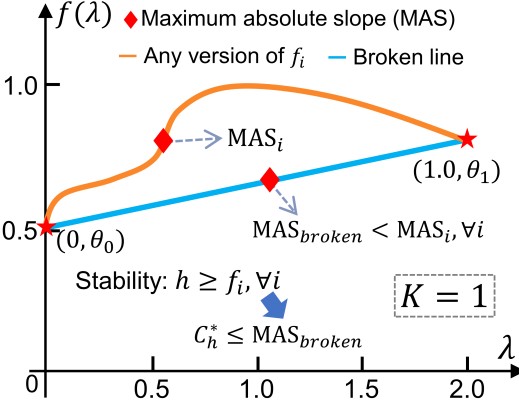

Figure 6: The way to find upper bound of $C_h^*$, a case to explain Proposition 4.1.

**The relationship between $f(\lambda)$ and $\{\theta_k\}_{k=0}^K$.** Setting the polynomial order to $K$, our model learns a set of $K + 1$ weights $\{\theta_k\}_{k=0}^K$ that acts as the coefficients for polynomial approximation, such that $f(\frac{2k}{K}) = \theta_k$, for all $k \in [0, K] \cap \mathbb{Z}$. Put differently, any 2-D curve passing through all points of $(0/K, \theta_0), (2/K, \theta_1)..,(2K/K, \theta_K)$ can be regarded as a possible version of $f(\lambda)$ for polynomial approximation. Consequently, given any set of $\{\theta_k\}_{k=0}^K$, the stability of $h(\lambda)$ is not worse than the most stable version among all possible $f(\lambda)$.

Formally, we naturally extend the Theorem 4.1 to the following proposition, which provides a theoretical upper bound for $C_h^*$ of $h(\lambda)$, directly using the learnable coefficients $\{\theta_k\}_{k=0}^K$.

As $h$ is at least as stable as the most stable $f$, the MAS of $h$ can be upper bounded by the minimum MAS among all possible $f$ functions, which is infinitely close to the MAS of the broken line passing through points $\{(k/K, \theta_k)\}_{k=0}^K$. Figure 6 helps to better understand Proposition 4.1. For example, let us set $K$ to 1, any curve passing through $(0, \theta_0)$ and $(2, \theta_1)$ can be a version of $f$. Out of all possible $f$ versions, the line segment directly connecting two points has the minimum MAS (i.e., $\frac{|\theta_0 - \theta_1|}{2}$).

We have introduced in the main text the relationship between $f(\lambda)$ and $\{\theta\}_{k=0}^K$. Based on this, the proof of Proposition 3.2 is equivalent to proving the following lemma.

**Lemma B.1.** *Given $K + 1$ points $(0, \theta_0)$, $(\frac{2}{K}, \theta_1)$, ..., $(\frac{2K}{K}, \theta_K)$, suppose there are infinitely many functions $f(\lambda)$ that pass through these $K + 1$ points. The maximum absolute slope (MAS) of any version of a function $f(\lambda)$ is not less than the MAS of the piecewise linear function passing through these $K + 1$ points.*

*Proof.* Let $f_{broken}(\lambda)$ be the piecewise linear function passing through the given $K + 1$ points. The slope of $f_{broken}(\lambda)$ between two consecutive points $(\frac{2i}{K}, \theta_i)$ and $(\frac{2(i+1)}{K}, \theta_{i+1})$ is given by:

$$m_i = \frac{\theta_{i+1} - \theta_i}{\frac{2(i+1)}{K} - \frac{2i}{K}} = \frac{\theta_{i+1} - \theta_i}{\frac{2}{K}} \tag{25}$$

The MAS of $f_{broken}(\lambda)$ is the maximum of the absolute values of these slopes:

$$\text{MAS}_{broken} = \max_{0 \le i \le K-1} |m_i| \tag{26}$$

Now, consider any function $f(\lambda)$ that passes through the $K + 1$ points. Since $f(\lambda)$ is differentiable, by the mean value theorem, for each interval $\left[\frac{2i}{K}, \frac{2(i+1)}{K}\right]$, there exists a point $\lambda_i$ such that:

$$f'(\lambda_i) = \frac{f\left(\frac{2(i+1)}{K}\right) - f\left(\frac{2i}{K}\right)}{\frac{2(i+1)}{K} - \frac{2i}{K}} = \frac{\theta_{i+1} - \theta_i}{\frac{2}{K}} = m_i \tag{27}$$

Since $f(\lambda)$ passes through all the given points, we have:

$$\text{MAS}_f \ge \max_{0 \le i \le K-1} |f'(\lambda_i)| = \max_{0 \le i \le K-1} |m_i| = \text{MAS}_{broken} \tag{28}$$

Thus, the MAS of any function $f(\lambda)$ satisfying the conditions is greater than or equal to the MAS of the piecewise linear function $f_{broken}(\lambda)$ passing through the points. $\square$

## C Results on DIPS

Table 3 shows the results on DIPS. Kindly note that we train and test all methods both on the DIPS dataset.

Table 3: **Training and testing on DIPS.**

| Methods | Native-Bound | ESMFold |
|---|---|---|
| BIPSPI [36] | 0.891 (0.003) | 0.883 (0.012) |
| SASNet [43] | 0.895 (0.002) | 0.878 (0.013) |
| dMaSIF [39] | 0.909 (0.012) | 0.885 (0.004) |
| DTNN [37] | 0.902 (0.009) | 0.870 (0.014) |
| NEA [15] | **0.911 (0.015)** | 0.875 (0.009) |
| Ours w/o SR | 0.909 (0.013) | 0.882 (0.013) |
| **Ours** | 0.909 (0.007) | **0.906 (0.007)** |

## D The cross-attention layer used

$$\boldsymbol{H}_1' = \text{softmax}\left(\frac{(\boldsymbol{H}_1\boldsymbol{W}_Q)(\boldsymbol{H}_2\boldsymbol{W}_K)^T}{\sqrt{d}}\right)(\boldsymbol{H}_2\boldsymbol{W}_V), \tag{29}$$

$$\boldsymbol{H}_2' = \text{softmax}\left(\frac{(\boldsymbol{H}_2\boldsymbol{W}_Q)(\boldsymbol{H}_1\boldsymbol{W}_K)^T}{\sqrt{d}}\right)(\boldsymbol{H}_1\boldsymbol{W}_V). \tag{30}$$

## E Hyper-parameters

The training process takes around 0.5 hours with 1 Nvidia 4090 GPUs with 24GB RAM. The hyper-parameters used in this paper are listed in the following table.

| Hyperparameters | Values |
|---|---|
| Graph node degree (k-NN) | 10 |
| Filter coefficient number for BernNet | 10 |
| Dimension of $\text{MLP}_e$ in Eq. 5 | 27, 1 |
| Number of layers in BernNet | 3 |
| Dropout rate | 0.2 |
| Number of attention head | 4 |
| Weight of loss $\mathcal{L}_I$ | 1.0 |
| Weight of loss $\mathcal{L}_S$ with BernNet | 0.35 |
| Batch size | 4 |
| Learning rate | $3 \times 10^{-4}$ |
| Optimizer | Adam |

Table 4: Hyperparameter choices of ATPROT.

