# OpenReview forum: "Towards Stable Representations for Protein Interface Prediction"
_NeurIPS.cc/2024/Conference — NeurIPS 2024 poster_

### Official Review · Reviewer_RNrt · 2024-07-02

**Soundness:** 3
**Presentation:** 4
**Contribution:** 3
**Rating:** 6
**Confidence:** 5

**Summary:**

This work focuses on protein interface prediction, i.e., determining whether a pair of residues from different proteins interact. It notices the conformational change within the protein upon binding and regards the flexibility as an attack on the trained model. An adversarial training framework named ATProt is introduced to defend against this kind of attack with a theoretical guarantee. Experiments demonstrate consistent improvements. However, I am not completely certain about many unignorable points, such as the experimental setting, the missing of up-to-date baselines, and etc. I am absolutely fond of the idea of being robust to conformational changes and give a score between 4 and 5 (leave the decision to other reviewers and AC).

**Strengths:**

(1) This paper pinpoints an important topic in computational biology, that is, the domain shift for bounded and unbounded structures when evaluating deep learning models. It is a real-world dilemma that scientists do not hold bound-form structures for new proteins.

(2) ATProt is proposed to achieve stability-regularization. It has a theoretically upper bound associated with the Bernstein basis and a stability-oriented objective is introduced. The overall framework can be generalized to many sorts of GNNs, which is also a big strength.

(3) The author conducted many different experimental settings, including native unbound structures, and structures produced by AF2, and ESMFold. The results showcase the superiority of ATProt and demonstrate the decrease of DL models when using predicted or native structures as inputs.

**Weaknesses:**

(1) The problem formulation is kind of confusing. The authors target classifying all possible pairs of residues from separate proteins. I suppose this may not be the regular version of protein interface prediction (PIP). The standard one would be to classify whether each residue on the ligand and receptor is on the interface or not. The PIP task in this paper is more challenging than the standard PIP task, and is very close to (but easier than) the protein-protein docking problem. Because once the pairwise interactions are recognized, the docking pose is nearly known. I recommend the authors reconsider the problem definition and formulate it more carefully.

(2) No adequate ablation study. I believe a more proper ablation analysis is needed to showcase the contribution of each component. For instance, it would be interesting to see ATProt-SGC/CHEBY/LPF's performance without the stability regularization.

(3) More baselines are required. I notice that all baselines are pretty old, containing NEA (2017), dMasif (2021), BIPSPI (2019), SASNet (2019), DTNN (2017), and DCNN (2016). It is not convincing enough to select these old-school algorithms as competitors. There has been a surge of interest in developing DL models for interface identification. Please see below. Moreover, it is necessary to also evaluate ESM-Fold and Alphafold-2 (if possibly, AF3) in this problem, because they can also immediately output the complex structures (so the interface is given) from sequences. These structure prediction models completely do not suffer from structural perturbation.

[A] Epitope 3D: a machine learning method for conformational b-cell epitope prediction. Brief in Bioinformatics, 2022.

[B] ScanNet: an interpretable geometric deep learning model for structure-based protein binding site prediction, Nature Methods, 2022.

[C] PesTo: parameter-free geometric deep learning for accurate prediction of protein binding interfaces. Nature Communications, 2023.

[D] SEMA-2.0: Antigen b-cell conformational epitope prediction using deep transfer learning. Frontiers in Immunology, 2022

[E] Pair-EGRETt: enhancing the prediction of protein-protein interaction sites through graph attention networks and protein language models. ICLR Workshop, 2023.

(4) I really appreciate the authors' efforts in attacking the structure change's bad effect in DL models. However, there is an unavoidable issue that I want to discuss with the authors and AC. If our concentration is to bridge the gap between training (bound) and test (native/unbound) datasets, why does not the author directly train the model on unbound or predicted structures? To be specific, we can use AF2/ESMFold to obtain unbound structures and leverage them as the training data. Moreover, it is also feasible to rely on Rosetta, PyMol, OpenMM, etc. to relax the bound structures with some force fields and approximate the unbound circumstance. In the AI4S community, I believe the solution is more flexible and we should consider more simple but effective methods to solve the problem. There have already been some works [A] that mention this representation shift between real structures and predicted structures.

[A] Protein 3D Graph Structure Learning for Robust Structure-Based Protein Property Prediction. AAAI 2024.

**Questions:**

(1) In Figure 3, how do the authors increase the structure change of test samples? I agree with Equ. 5 to use \Delta X to represent the structure perturbation. However, I suppose there are some constraint in this \Delta X. Notably, the protein flexibility is not entirely random. The atoms' movement need to follow some physical rules. See [A] for illustration.

[A] Fractional denoising for 3d molecular pre-training. ICML 2023.

(2) As far as I am concerned, dMasif is a surface point cloud-based mechanism, and it only outputs the prediction of surface points instead of residues. How do the authors make the comparison compatible?

(3) The conformational change is fancy without doubt and many studies have been trying to do relevant analysis. For instance, ProtMD [A] pretrains the DL model on MD data to learn the protein flexibility with continuous time domain. Can the authors do an additional experiment to see whether this category of pretraining is effective to mitigate the negative effect of conformation change for PIP?

[A] Pre-training Protein Geometric Models via Molecular Dynamics Trajectories. Advanced Science, 2022.


(4) It looks weird to me that the pretraining on DIPS brings a negative impact on dMasif but positive impacts for all the others. After pretraining, dMasif drops from 0.928 to 0.922 for native-bound, from 0.912 to 0.903 for native-unbound, etc. Can the authors explain this counterintuitive phenomenon?

**Limitations:**

Yes, the limitation is addressed.

---

> ### Author Rebuttal · Authors · 2024-08-07
>
> We sincerely appreciate the reviewer's thorough review and insightful feedbacks. We also appreciate the reviewer's recognition for the core idea and contribution of this paper. We will respond to these comments point-by-point.
>
> **[Cons. 1 Problem Formulation]**
> Thank you for your professional comments, which we fully agree with. We will make some adjustments to avoid confusion. To address your concerns, we make the following efforts.
>
> 1. Firstly, we clarify that existing research related to our task falls into three categories: partner-specific interface prediction (PS-IP), partner-specific binding site prediction (PS-BSP), partner-agnostic binding site prediction (PA-BSP).
>
>     **PS-IP** is the problem of our paper, which is to predict whether there is an interaction between a pair of residues of two different proteins. In many cases (e.g., NEA, SASNet, BIPSPI, HOPI), this task is referred to as 'protein interface prediction'.
>
>     **PS-BSP**, as mentioned in your comment, predicts whether the residues belong to the interface. For this task, the interface of a protein is conditioned on its partner.
>
>     **PA-BSP** predicts all possible binding sites of a protein. It is the main setting of ScanNet and PesTo, which does not assume knowledge of the partner.
>
> 2. Although the applications of the three tasks differ, they are strongly related. We will mention the above three tasks and their differences in the abstract and introduction sections. Importantly, in the related work section, we will provide descriptions of baseline methods belonging to each.
>
> **[Cons. 2 More Ablation Study]**  Thanks for the valuable suggestion. To address the your concerns, we show ablation results on stability regularization (SR) in Table 3,4 in the one-page PDF.
>
> We find that:
>
> (1) In the Native-Bound scenario, SR is ineffective, while in the remaining (real-world) scenarios, SR significantly improves all four cases.
>
> (2) The gain of SR on the SGC encoder is relatively small among the four cases, which may be due to the direct reduction of the polynomial order $K$ in SGC. Therefore, the expressive power is explicitly diminished.
>
> **[Cons. 3 More Baselines]**
> Thanks for the constructive feedback. Regarding the five papers provided, we find that except for Pair-EGRET, they are all designed for the task of 'partner-agnostic binding site prediction'. However, we find that PesTo can perform the PIP task after simple code modifications.
>
> **In this way, we consider Pair-EGRET, PesTo, ESM-Fold and AF2 as additional baselines.**
>
> The results are shown in Table 7 in the one-page PDF. We find that ATProt can still outperform AF2 (actually AF-Multimer) and ESM-Fold.
>
> **[Cons. 4.1 Use AF2 to Generate Training Data]**
> This is a very worthy topic for discussion. Frankly speaking, this is our initial attempt. Table 7 in one-page PDF shows the results. The results indicate that using the suggested methods is difficult to achieve SOTA.
>
> Regarding this, our insights are:
>
> 1. There may still be distribution shift between the training and testing sets. This is because the cropping process of AF2 considers monomers and multimers. This results in AF2's training set being a mixture of bound and unbound structures. Thus, using AF2 to generate both training and test sets may not necessarily guarantee consistent distribution.
>
> 2. It is not very practical to use AF2 to generate training structures on a large-scale dataset (e.g., DIPS with over 40,000 dimers), as it would be very time-consuming.
>
> 3. Our method allows **training only once** and then testing on **various versions of structures**. This uniformity and wide applicability are also a potential advantage.
>
> **[Cons. 4.2 Relax the Bound Structures]**
> Performing force field optimization based on First Principles is a very insightful idea and has been proven effective[1]. However, molecular dynamics (MD) often requires huge computational cost. Thus, applying MD to large-scale datasets is a challenge.
>
> [1] Benchmarking Refined and Unrefined AlphaFold2 Structures for Hit Discovery. ACS, 2023.
>
> **[Cons. 4.3 Representation Shift]**
> Thanks for providing this literature called SAO. SAO indicates that there is room for improvement when training and testing both on AF2-structures (i.e., its TonP baseline). Although tasks are different, the deep motivations of ATProt and SAO are similar: both to enable effective inference using structures from various source.
>
> **[Q1. How to Change Structures]** We strongly agree with your point that flexibility cannot be arbitrarily generated. We applied a dynamics guided method called ProDy. It can sample structures that differ from the initial structure by a specific value (RMSD).
>
> **[Q2. About dMaSIF]** The main point of dMaSIF is to generate point cloud representations. However, the dMaSIF paper also contributes a geometric convolutional model, enabling it to address structure-related downstream tasks. Importantly, the PIP task is implemented in the dMaSIF paper.
>
> **[Q3. ProtMD]**
> Thanks for providing this valuable literature, which has a user-friendly open-source nature.
>
> In Table 7 of the one-page PDF, we present the experimental results of ProtMD. We consider the results of fine-tuning (ProtMD-FT) and linear probing (ProtMD-LP). Surprisingly, its performance slightly exceeds the baseline specifically designed for PIP.
>
>
> **[Q4. Pre-training with dMaSIF]**
> Thanks for the thorough findings. The quality of the DIPS and DB5.5 dataset differs slightly. The latter is an expert-annotated "gold standard dataset", while the dimers mined in DIPS are mainly from multimers in PDB, which do not reflect pure **binary interaction knowledge**. Facing such gap, negative transfer (where pre-training has a negative impact) is common for some models because they are not designed for the pretraining-finetuning paradigm.

---

> > ### Comment · Reviewer_RNrt · 2024-08-08
> > **Update**
> >
> > Thanks for the authors' efforts in answering my question. I am glad to see that ATProt-Bern outperforms all essential baselines and stands the No. 1 in this binding site prediction task. I am also happy to know that my suggestion helps the author clarify more clearly the concepts and definition of interface v.s. binding site.
> >
> > I understand that a conference paper cannot totally solve the binding site prediction task and there is still a long way to go, e.g., considering more training data (like using Alphafold Database), using MD simulations to relax the structures, designing domain adaptation and generalization algorithms to adapt the methods to predicted structures, etc. Based on this fact, I believe ATProt-Bern is worthy of publication and should be regarded as an adequate contribution to this important challenge.
> >
> > I have raised my score to 6 and recommend the authors to incorporate the updates to the final revisions, such as the ablation studies, and the new baseline (PesTo, ProtMD).
> >
> > Last but not least, I am still confused by this reply: "[Q2. About dMaSIF] The main point of dMaSIF is to generate point cloud representations. However, the dMaSIF paper also contributes a geometric convolutional model, enabling it to address structure-related downstream tasks. Importantly, the PIP task is implemented in the dMaSIF paper."
> >
> > As far as I am concerned, the quasi-geodesic convolutional model is performed in the point cloud instead of the structure. My question is how the author mapped the prediction of dMaSIF on the surface point to the residues?

---

> > > ### Author Response · Authors · 2024-08-09
> > > **Thank you for raising the score and helping us improve.**
> > >
> > > Thank you for raising the score. We greatly appreciate your recognition and assistance of our work.
> > >
> > > Based on your constructive suggestions, we commit to including all the added baselines in the final version manuscript. Additionally, we will introduce a discussion on AF2 data augmentation (providing experimental results) and the use of MD for relaxation.
> > >
> > > ***About dMaSIF***
> > >
> > > We previously misunderstood your question. Please allow us to explain again. Regarding dMaSIF, we did perform additional processing to make it compatible with the PIP task. In the dMaSIF paper, the predicted elements of the interface are **not residues but points** simulated by the dMaSIF method. This can be found in its source code and paper, for example: *"For interaction prediction, we compute dot products between the feature vectors of both proteins to use them as interaction scores between pairs of points."* This setting is obviously different from the mainstream approach of PIP, as it calculates the link probability between points rather than residues. Additionally, when we reproduced dMaSIF under its default setting, the result was only around 0.85, indicating significantly lower performance.
> > >
> > > **Therefore, we made modifications by referencing the processing manner in the ScanNet paper.**
> > >
> > > In the ScanNet paper, to allow for a fair comparison with MaSIF-site, they converted MaSIF's predictions from the surface point level to the residue level. This is described in the ```Baseline methods/masif-site``` section of ScanNet. The approach is to first assign each point to the nearest atom and the corresponding residue. Then for each residue, take the highest probability among all its points as the final result. However, things are slightly different because we need to calculate the residue-residue link probability rather than classify each residue in the binding site prediction task.
> > >
> > > **Therefore, we made some minor adjustments:**
> > >
> > > We first assign each point to the nearest atom and the corresponding residue. Then for each **residue-residue pair**, take the highest probability among all its point pairs as the final result. For an illustrative example, let's assume two proteins have 1 and 2 residues respectively, labeled as {1} and {1',2'}. dMaSIF generates a total of 6 surface points for these 3 residues. The correspondence between points and residues is: {a,b$\rightarrow$1}, {c,d$\rightarrow$1'}, {e,f$\rightarrow$2'}. dMaSIF can outputs the linking probabilities between points: {a,c:0.2},{a,d:0.4},{a,e:0.6},{a,f:0.8},{b,c:0.1},{b,d:0.3},{b,e:0.5},{b,f:0.7}. Finally, for residue pair {1,1'}, we pick the highest value among {a,c:0.2},{a,d:0.4},{b,c:0.1},{b,d:0.3} (i.e., 0.4) as the result. For residue pair {1,2'}, we pick the highest value among {a,e:0.6},{a,f:0.8},{b,e:0.5},{b,f:0.7} (i.e., 0.8) as the result. We tried alternative approaches for ```pick the highest value```, such as taking the average of all relevant probabilities or the average of the top k probabilities, but they were not as effective as simply using the highest value.
> > >
> > > Thus, we can reformulate dMaSIF into our PIP setting. Under this setting, its performance can match or even surpass that of several baselines.

---

### Official Review · Reviewer_RLcQ · 2024-07-12

**Soundness:** 3
**Presentation:** 2
**Contribution:** 3
**Rating:** 7
**Confidence:** 4

**Summary:**

This paper first identifies a commonly overlooked issue in protein docking: protein flexibility, and proposes an improved method to address it. This approach utilizes an adversarial training framework to maximize Lipschitz continuity. Experimental results demonstrate the effectiveness of this method.

**Strengths:**

The overall quality of this paper is very high. The addressed issue - protein flexibility - is highly meaningful. The methods are sound. Therefore, I believe this paper should be accepted.

**Weaknesses:**

The writing in one part of the paper is not very clear. I hope the authors can adjust it. The paper discusses a method of adversarial training to ensure Lipschitz continuity, but the specific training objectives of adversarial training are not explicitly stated here. Adversarial training typically refers to having two competing objectives, which are contradictory. However, I did not see this aspect described in the paper.

**Questions:**

None

**Limitations:**

Excessive emphasis on ensuring Lipschitz continuity may limit the expressive power of the network ($L_S$). The article is meaningful and interesting. However, excessive restriction on the network's expressive ability could potentially prevent this method from becoming the ultimate solution. Therefore, I believe this article does not reach the level of spotlight or oral presentation.

---

> ### Comment · Reviewer_RLcQ · 2024-07-31
>
> Let me add that I believe the method proposed by the author is a form of regularization rather than an adversarial training approach. If I am correct, please adjust the relevant writing accordingly.

---

> ### Comment · Reviewer_RLcQ · 2024-08-05
>
> Also, please add the ablation study and baselines (e.g. EBMdock) as reviewer RNrt mentioned.

---

> ### Author Rebuttal · Authors · 2024-08-07
>
> We thank the reviewer **RLcQ** for acknowledging our method as "meaningful and interesting", the quality of our paper as "very high" and our methods as "sound". Many thanks for the constructive feedback especially for the questions about the writing part. These all help us to further enhance the readability and the quality of our paper.
>
> **[Cons. Description of Adversarial Training]**
>
> ***Our explanation:***
>
> Thanks for the constructive feedbacks. We strongly agree with you that adversarial training (AT) typically involves two or more conflicting training objectives. Many cases of literature indicate that a more accurate statement is: Lipschitz continuity regularization is a popular and effective method for **Adversarial Robustness (AR)**.
>
> From the formulation of the training objectives, two losses only exhibit intuitive conflicts, as $\mathcal{L}_S$ acts by constraining the slope of the GNN filter, which limits the expressive power of the GNN (as mentioned in the Limitation section). Moreover, from the empirical perspective, the regularization loss slightly reduces the model's performance on the 'Native-Bound' structure, implying that there may be an **implicit conflict** in the training objective. Therefore, we acknowledge that the two objectives do not have explicit or theoretically provable conflicts.
>
> In our work, robustness is the model's performance when facing such flexibility 'attack', that is, when we test on unbound, ESM-Fold or AF2 structures. Therefore, while our training objectives ($\mathcal{L}_I$ and $\mathcal{L}_S$) have implicit conflicting nature, we believe that AR will be a more conducive expression for promoting readability.
>
> ***Our adjustment:***
>
> 1. Firstly, as shown in Table 3,4 in one-page PDF, we note that the proposed regularization slightly impairs performance in the standard, i.e., Native-Bound scenario. Thus, we validate the conflicting nature of the objectives in the empirical perspective.
>
> 2. Additionally, we will supplement the formulation analysis of the objective conflicts, i.e., how constraining the slope of the GNN filter affects its expressive power.
>
> 3. We will adopt the use of "adversarial robustness" instead of "adversarial training" which will involve the Preliminaries section. As a result, the training objective for protein representation stability will have a more rigorous description.
>
> **[Cons. Ablation Study]**
> Thanks for the valuable suggestions.
>
> ***About baselines:***
>
> We have added various baselines, and all results can be found in Table 7 in the one-page PDF. We have added the following baselines: **EBMdock, Pair-EGRET, PesTo, ProtMD-FT, ProtMD-LP, ESM-Fold, AF2 and AF2-Train**.
>
> ***About regularization:***
>
> We perform ablation study on the effectiveness of the stability regularization. The results are shown in Table 3,4 in the one-page PDF.
>
> **We will include all newly added experiments to our revised manuscript**

---

> ### Comment · Area_Chair_qiio · 2024-08-13
>
> Dear RLcQ,
>
> Have you had the chance to read the authors’ rebuttal, and does it address the concerns you raised, and has it influenced your evaluation of the paper?
>
> Best regards, AC

---

> > ### Comment · Reviewer_RLcQ · 2024-08-13
> >
> > Dear AC,
> >
> > I have reviewed the authors' response. I have no further questions and stand by my original assessment. I believe the paper has reached the standard for acceptance, but it does not meet the criteria for spotlight or oral presentation.

---

### Official Review · Reviewer_kBXn · 2024-07-13

**Soundness:** 3
**Presentation:** 2
**Contribution:** 3
**Rating:** 6
**Confidence:** 4

**Summary:**

This paper considers the generalization issue caused by the conformational changes of two proteins before and after binding in the PIP task. The authors view protein flexibility as an attack on the model and aim to defend against it for better generalization. Hence, an adversarial training framework for protein representation is proposed, termed ATProt. ATProt is theoretically proven to ensure protein representation stability under complex protein flexibility. Experiments on the DB5.5 and DIPS datasets further validate the effectiveness of ATProt, especially when using unbound structures as input.

**Strengths:**

* The concept of viewing protein flexibility as an attack on the model and utilizing an adversarial training framework to address this issue is both interesting and brilliant.
* Experiments on the DB5.5 and DIPS datasets effectively validate ATProt's performance improvement in unbound scenarios.
* ATProt provides a novel perspective for robust predictions with real unbound inputs. Combining ATProt with protein structure prediction models such as AlphaFold and ESMFold might aid in advancing sequence-based PIP tasks.

**Weaknesses:**

* Although the idea of ATProt is interesting and effective, the model's architecture adopts a combination of existing methods, BernNet and Cross-attention.
* The ablation experiment targeting 'SR' was conducted only within the 'ATProt-BearNet' combination.
* Lacking some visualization experiments. No examples are presented to show which residue pairs are predicted more accurately before and after incorporating ATProt.
* In Supplementary.E, some hyperparameters are missing, such as the number of epochs and the number of Cross-attention layers.

**Questions:**

* What about the experimental results for the removal of 'SR' in the other three encoders? Clearly demonstrating the performance improvement brought by 'SR' in the other three encoders would better validate the effectiveness of the framework.
* Considering visualizing the changes in interface prediction before and after adding 'SR' could more intuitively validate the proposed framework..
* Showcasing the hyperparameter tuning process for “the weight of loss $L_{s}$ with BernNet” would be beneficial, as it is the core idea of ATProt.
* Providing the number of epochs and the number of cross-attention layers would help readers better reproduce the results.

**Limitations:**

The authors have thoughtfully addressed the limitations of their study, particularly highlighting the absence of adversarial training-based classifiers in their framework.

---

> ### Author Rebuttal · Authors · 2024-08-07
>
> We sincerely appreciate the reviewer **kBXn**'s recognition of our paper and valuable comments. We will respond to reviewer's insightful suggestions point-by-point.
>
> **[Cons 1. Novelty of Model's Architecture]**
> Thanks for the valuable comment. We sincerely clarify that the novelty of this paper is three-fold.
>
> 1. **Various protein graph encoders:** We fully agree with your feedback that we did not pay enough attention to the design of new protein representation models. However, **the proposed method (strategy) is plug-and-play**. Although we only demonstrate four cases, the GNN encoders applied for protein-related tasks are far more numerous [1,2]. The concept of stability regularization (SR) has the potential to be widely applied to **various existing encoders** to address the challenges posed by protein flexibility.
>
> 2. **Various protein-related tasks:** Through empirical and theoretical validation, we show that in the protein interface prediction (PIP) task, we can avoid simulating the exact distribution of protein flexibility and instead use **simple yet effective regularizations**. In other protein-related tasks, this strategy might also address distribution shifts caused by structural changes. For example, SR has the potential to resolve the distribution shift between AlphaFold 2 and native structures mentioned in [3].
>
> 3. **New proposition for stable BernNet:** BernNet is a widely applied model with strong expressive power. Here, we introduce the Lipschitz regularization for BernNet for the first time (to our knowledge), achieving state-of-the-art performance on the PIP task. Due to the analytical difficulties caused by the complex polynomial structure of BernNet, we respectfully believe it is a relatively fundamental contribution to the field of stable GNNs.
>
> **[1]** Protein representation learning by geometric structure pretraining. ICLR, 2023.
>
> **[2]** Structure-based protein function prediction using graph convolutional networks. Nature communications, 2021.
>
> **[3]** SaProt: Protein language modeling with structure-aware vocabulary. ICLR, 2024.
>
> **[Cons 2. Need More Ablation Experiments]**
> We really appreciate the reviewer's detailed comments. We include the added experimental results in Table 3,4 in one-page PDF.
>
> From the results, we can see that:
>
> (1) In the Native-Bound (ideal and virtual) scenario, the proposed SR is ineffective, while in the remaining (real-world) scenarios, SR significantly improves all four cases of encoders.
>
> Furthermore, we have made a new discovery that:
>
> (2) The gain of SR on the SGC encoder is relatively small among the four cases, which may be due to the direct reduction of the polynomial order $K$ in SGC caused by SR. Therefore, the expressive power is explicitly diminished.
>
> **We will add all of these results to our revised manusrcipt.**
>
> **[Cons 3. Residue-Level Visualizations]**
> Thank you for your insightful comments regarding this issue. To address your concern, we add the residue-level visualization, and the figure is included in one-page PDF. Due to space constraints, we select two out of the 25 samples from the DB5.5 testset.
>
> The baseline method we select is NEA in our manuscript. First, we categorize the misclassified samples by NEA into **false negatives (FN)** and **false positives (FP)**. Due to the imbalanced nature of the PIP problem, we observe that there are notably more FP samples than FN samples. Therefore, in the added figure, we visualize all FN samples and some of the FP samples. Please kindly note that all of the visualized pair samples are misclassified by NEA but correctly classified by ATProt.
>
> From the figure we add, it can be seen that the residues involved in the shown samples (especially the FN samples), exhibit significant structural changes.
>
> In addition, we quantify the average structural changes of residues corresponding to correct and incorrect predictions for both NEA and ATProt. This is performed with all of the DB5.5 testset samples. For NEA, the average structural changes of correctly and incorrectly classified residues are 0.563 and 0.972, respectively. For ATProt, the respective values are 0.589 and 0.597. In simple terms, the structural changes impair the performance of NEA but have little impact on ATProt.
>
> This insightful comment highlights the advantages of the proposed method, and **we will include this interesting part in our revised manuscript**.
>
> **[Cons 4. Some Hyperparameters are Missing]**
> Thanks for your very thorough review. Table 5 in one-page PDF is a comprehensive summary of the hyperparameters. The **bolded** rows are newly added.
>
> **[Question 1. Effectiveness of SR]**
> Thanks for the feedback, and we have added more experimental results in the response to **[Cons 2]**.
>
> **[Question 2. The Prediction Changes after Using SR]**
> Thanks for the valuable comment. In the response to **[Cons 3]**, we conducted visualizations and quantitative experiments, both of which show the robustness of ATProt to structural changes.
>
> **[Question 3. Hyperparameter Tuning Process for Weight of $\mathcal{L}_S$]**
> Thank you very much for this question. For the overall training objective $\beta\mathcal{L}_I + \gamma\mathcal{L}_S$, we fix $\beta$ at $1.0$ and search for $\gamma$ in the range of $0.1$ to $10$. For fairness, we ensure that all baselines and our methods conduct hyperparameter search with ```n_trials=50``` in the Optuna API.
>
> We provide the results on the DB5.5 dataset in Table 6 in one-page PDF.
>
> We observe that all three cases exhibit a similar pattern: with the increase of $\gamma$, **the performance first increases and then decreases**. The difference is that the optimal weight ranges for the three types of encoders are not the same.
>
> **[Question 4. More Detailed Hyperparameters]**
> Thanks for your detailed suggestion, and we have provided more hyperparameter results in the response to **[Cons 4]**. The complete hyperparameter table will be added to our revised appendix.

---

> ### Comment · Area_Chair_qiio · 2024-08-13
>
> Dear kBXn,
>
> Have you had the chance to read the authors’ rebuttal, and does it address the concerns you raised, and has it influenced your evaluation of the paper?
>
> Best regards,
> AC

---

### Author Rebuttal · Authors · 2024-08-07

Dear reviewers,

We sincerely appreciate your valuable time and constructive feedbacks.

**Please see the attached one-page PDF with a summary of added experimental results.** It contains:

Figure 6: Visualization of the changes in interface prediction results before and after using stable regularization (SR).

Table 3,4: The ablation study on the effectiveness of SR.

Table 5: Hyperparameter choices.

Table 6: The tuning process of the weight of $\mathcal{L}_S$.

Table 7: Results of the newly added baselines (EBMDock, Pair-EGRET, PesTo, ProtMD-FT, ProtMD-LP, ESM-Fold, AF2 and AF2-Train). In this table, to address the concern of **Reviewer RNrt**, AF2-Train represents using the training structures generated by the AF2.

Please see our reviewer-specific feedback for more information.

---

### Decision · Program_Chairs · 2024-09-25

**Decision:**

Accept (poster)

**Comment:**

The paper proposes ATProt, an adversarial training framework for protein interface prediction, designed to enhance model robustness against protein flexibility by treating it as an attack, with experiments showing improved performance across various benchmarks. Reviewers appreciated the novel approach of treating protein flexibility as an adversarial attack, the theoretical grounding and soundness of the framework, and its demonstrated robustness across different datasets. Reviewers criticized the lack of clarity in the problem formulation, the limited novelty in the model architecture, inadequate ablation studies, missing up-to-date baselines, and the need for more detailed experimental settings and visualizations to substantiate better the claims made in the paper. The authors addressed many of these issues during the discussion period, and subsequently, the reviewers unanimously recommended that the paper be accepted. Therefore, we accept the paper, but we urge the authors to carefully incorporate all the additional updates and results in the final revision.